# Study of *Bacillus cereus* as an Effective Multi-Type A Trichothecene Inactivator

**DOI:** 10.3390/microorganisms12112236

**Published:** 2024-11-05

**Authors:** Fernando Abiram García-García, Eliseo Cristiani-Urbina, Liliana Morales-Barrera, Olga Nelly Rodríguez-Peña, Luis Barbo Hernández-Portilla, Jorge E. Campos, Cesar Mateo Flores-Ortíz

**Affiliations:** 1Laboratorio Nacional en Salud, Facultad de Estudios Superiores Iztacala, Universidad Nacional Autónoma de México, Av. de los Barrios No. 1, Tlalnepantla 54090, Mexico; abiramhiz@gmail.com (F.A.G.-G.); lbarbo@unam.mx (L.B.H.-P.); 2Departamento de Ingeniería Bioquímica, Escuela Nacional de Ciencias Biológicas, Instituto Politécnico Nacional, Av. Wilfrido Massieu s/n, Unidad Profesional Adolfo López Mateos, Ciudad de México 07738, Mexico; lilianamor@prodigy.net.mx; 3Laboratorio de Biogeoquímica, Unidad de Biología, Tecnología y Prototipos (UBIPRO), Facultad de Estudios Superiores Iztacala, Universidad Nacional Autónoma de México, Av. de los Barrios No. 1, Tlalnepantla 54090, Mexico; onel.rodriguez@iztacala.unam.mx; 4Laboratorio de Bioquímica Molecular, Unidad de Biología, Tecnología y Prototipos (UBIPRO), Facultad de Estudios Superiores Iztacala, Universidad Nacional Autónoma de México, Av. de los Barrios No. 1, Tlalnepantla 54090, Mexico; jcampos@unam.mx; 5Laboratorio de Fisiología Vegetal, Unidad de Biología, Tecnología y Prototipos (UBIPRO), Facultad de Estudios Superiores Iztacala, Universidad Nacional Autónoma de México, Av. de los Barrios No. 1, Tlalnepantla 54090, Mexico

**Keywords:** biological mycotoxin decontamination, diacetoxyscirpenol (DAS), HT-2 toxin, multi-mycotoxicosis, neosolaniol (NEO), proventriculus bacterium, T-2 toxin, type A trichothecenes

## Abstract

Type A trichothecenes are common mycotoxins in stored cereal grains, where co-contamination is likely to occur. Seeking new microbiological options capable of inactivating more than one type A trichothecene, this study aimed to analyze facultative anaerobe bacteria isolated from broiler proventriculus. For this purpose, type A trichothecenes were produced in vitro, and a facultative anaerobic bacterial consortium was obtained from a broiler’s proventriculus. Then, the most representative bacterial strains were purified, and trichothecene inactivating assays were performed. Finally, the isolate with the greatest capacity to remove all tested mycotoxins was selected for biosorption assays. The results showed that when the consortium was tested, neosolaniol (NEO) was the most degraded mycotoxin (64.55%; *p* = 0.008), followed by HT-2 toxin (HT-2) (22.96%; *p* = 0.008), and T-2 toxin (T-2) (20.84%; *p* = 0.014). All isolates were bacillus-shaped and Gram-positive, belonging to the *Bacillus* and *Lactobacillus* genera, of which *B. cereus* was found to remove T-2 (28.35%), HT-2 (32.84%), and NEO (27.14%), where biosorption accounted for 86.10% in T-2, 35.59% in HT-2, and 68.64% in NEO. This study is the first to prove the capacity of *B. cereus* as an effective inactivator and binder of multi-type A trichothecenes.

## 1. Introduction

Mycotoxins, defined as toxic secondary metabolites produced by certain filamentous fungi or molds [1], are found in mycelium or fungus spores, with the capacity to infest either crops in the field or post-harvest and representing a potential risk to animal and human health through intake of food and feed made from contaminated raw materials [2,3]. Around 500 mycotoxins have been recognized, and their number continues to increase [4]. However, only a few, such as aflatoxins, citrinin, deoxynivalenol (DON), ergotamine, ochratoxins, trichothecenes, zearalenone (ZEA), and others, represent a significant food safety challenge [5,6]. Trichothecenes are considered the largest group of mycotoxins (approximately 170), mainly produced as secondary metabolites by *Fusarium* species [7]. They are characterized by having a 12,13-epoxytrichothec-9-ene structure, where the epoxy group constitutes the active site leading to oxidative radical formation [8], and according to their epoxy ring substitution pattern, they are classified into four groups (Types A, B, C and D) [9]. Type A trichothecenes are common contaminants in cereal grain and due to their high toxicity are considered of greatest concern to food and feed safety [10], mainly including diacetoxyscirpenol (DAS), T-2 toxin (T-2), HT-2 toxin (HT-2), and neosolaniol (NEO) [11]. Since animals and humans consume mixed diets of varied origins, co-exposure to different trichothecenes has been documented [7,12]. Thus, multi-trichothecene contamination highlights a need to investigate single sources that may contain different trichothecenes.

To mitigate mycotoxicosis, three main detoxification strategies have been developed to treat contaminated grain, known as physical, chemical, and biological methods [13]. Within the biological methods, crude and purified enzymes from different biological sources have been proposed to modify the mycotoxin into products of lower toxicity or no toxicity when compared to the original toxin [14,15]. In addition, complete microorganisms such as bacteria, filamentous fungi, and yeasts isolated from soil, mycotoxin-contaminated products, and the digestive tract of animals [16] have also been used to biosorb or biotransform mycotoxins due to the enzymes they contain [17,18,19]. A large number of strains have been found to degrade mycotoxins [20], and some have been found to degrade more than one simultaneously [21,22,23,24,25]. However, regarding the study of trichothecene removal, strains have mostly been developed considering Type B [26,27,28,29,30,31] and, to a small extent, Type A trichothecenes [21,32,33].

Therefore, given that type A trichothecene contamination is usually mixed, and to further identify effective strains capable of removing a wider spectrum of mycotoxins, this study aimed to identify bacterial strains coming from a broiler’s proventriculus, capable of removing more than one type A trichothecene. For this purpose, we first obtained DAS, T-2, HT-2, and NEO from *Fusarium sporotrichioides.* Then, the facultative anaerobe bacterial consortium from a broiler’s proventriculus was isolated and mycotoxin removal assays were performed. Next, from the original consortium, the five most representative bacterial strains were isolated and purified, their morphological structure characterized, and their 16S rDNA sequence analyzed to assess their taxonomic identification, and isolate removal assays were performed. Finally, the isolate with the greatest capacity to reduce the concentration of all the studied type A trichothecenes was selected to perform biochemical tests and biosorption assays.

## 2. Materials and Methods

### 2.1. Mycotoxin Production, Isolation, and Quantification

To produce type A trichothecenes in the laboratory, a pure *Fusarium sporotrichioides* strain (NRLL3299) was grown on potato dextrose agar (PDA). Particular traits such as micellar growth, morphological development of reproductive structures, and pigment release were evaluated following [34]. Trichothecenes were biosynthesized by submerged fermentation in Czapeck–Dox liquid broth [35]. Once the fungus had released the mycotoxins into the culture medium, extractions were carried out using an extracting phase of methanol (MeOH) and ethyl acetate (EtAc) HPLC grade (1:3 *v/v*). Extractions were collected at different culture ages: extract 1 = 77 days, extract 2 = 21 days, and extract 3 = 6 days, and stored in amber bottles at 4 °C. The presence of DAS, T2, HT2, and NEO was determined by thin layer chromatography (TLC) [35] and quantified by high pressure liquid chromatography-electrospray-time of flight-mass spectrometry (HPLC-ESI-TOF-MS) following García-García et al. [36], where a positive ion mode HPLC system was used, consisting of a vacuum degasser, autosampler, and binary pump (Infinity 1260, Agilent Technologies, Santa Clara, CA, USA), equipped with an Zorbax Eclipse Plus column (C-18 RRHD, 1.8 μm, 2.1 × 100 mm; Agilent, Santa Clara, CA, USA). The column temperature was maintained at 25 °C. The mobile phase consisted of 10 mM of ammonium acetate (A) and HPLC-grade acetonitrile (ACN) (B). The gradient began with 85% A and 15% B, then changed to 100% B at minute 40 and was maintained for 10 min. The flow rate was 0.15 mL/min. The column equilibration time was 14 min. The injection volume was 5 μL. HPLC was coupled to TOF/MS (time of flight-mass spectrometry; Agilent 66230B; Yishun, Singapore) with an electrospray interface. Operating conditions included a gas temperature at 250 °C, gas flow at 6 L/min, nebulizer pressure at 60 psig, shredder at 200 V, skimmer at 65 V, and OCT RF Vpp at 750 V. Data were acquired using a Mass Hunter Workstation LC/MS data acquisition with 6200 series software version B.06.01, build 6.0.633.10 (2012, Agilent Technologies, Santa Clara, CA, USA). The presence of each mycotoxin was determined using its molecular ion derivatized with ammonium ion NH_4_^+^ (M+ 18 uma): DAS (384.2 *m/z*), T-2 toxin (484.25 *m/z*), HT-2 toxin (442.24 *m/z*), and NEO (400.18 *m/z*).

### 2.2. Bacterial Consortium

#### 2.2.1. Broiler Proventriculus Facultative Anaerobe Bacteria

Six Ross broilers raised under standard production conditions and fed a sorghum soy-based diet were donated by the CEIEPAv (Center for Teaching, Research, and Extension in Poultry Production, using its acronym in Spanish). Broilers were sacrificed in accordance with PROY-NOM-194-SSA1-2000 [37]. Their gastrointestinal tracts were obtained under aseptic conditions following NOM-110-SSA1-1994 [38], stored in sterilized glass bottles (121 **°**C for 15 min), and transported frozen to the laboratory. Under laboratory conditions, a sterile field was generated to separate the proventriculus from the gizzard. The six proventriculi were weighted (mg) and separated into two groups (P1 and P2); in each group, cuts were made until 10 g was produced and placed in Erlenmeyer flasks (125 mL), for a duplicate and three repetitions. To obtain the bacterial consortium, samples were covered with 100 mL of brain–heart infusion (BHI) to promote bacterial growth for 48 h at 40 ± 1 °C using an anaerobiosis chamber (BD GasPak), to which two GasPak Ez Anaerobe Container System sachets were added to generate an anaerobic atmosphere with 13% CO_2_. After 48 h, a subculture was grown in Erlenmeyer flasks (50 mL) using Lactobacilli MRS broth (Difco BD, Sparks, MD, USA), a selective culture medium to only grow facultative anaerobe strains, for 24 h at 40 ± 1 °C in aeration conditions.

#### 2.2.2. Enrichment of the Bacterial Consortium and Removal of Mycotoxins

A batch enrichment culture was carried out for each mycotoxin (NEO, T-2, and HT-2), in which, without agitation and in an aerobic environment, the mycotoxins concentration was gradually increased (0.1, 0.2, 0.4, 0.8, and 1.6 μg/mL), incubated at 40 ± 1 °C for 24 h [21,32]. In all trials, the experimental treatment consisted of the toxin and the bacterial culture (100 µL) in MRS culture medium. Control 1 contained MRS medium + mycotoxin, control 2 MRS medium + live cells, and control 3 MRS medium. Since no differences were detected in controls 2 and 3, only control 1 is included in the analyses throughout this manuscript. Experiments were performed in triplicate. To quantify the residual mycotoxins, 8 mL of ethyl acetate (EtAc) was added, tubes were vortexed for 30 s, and the organic phase was recovered and evaporated to dryness under reduced pressure. All samples were resuspended in 1.5 mL of methanol (MeOH) to quantify residual mycotoxin using HPLC-ESI-TOF-MS (high-pressure liquid chromatography-electrospray-time of flight-mass spectrometry) following García-García et al. [36].

### 2.3. Bacterial Isolates

#### 2.3.1. Purification of Bacterial Isolates

From the original bacterial consortium, the five most representative strains (I1, I2, I3, I4, and I5) were selected based on their abundance and morphology through streak sowing, and 500 μL of the highest mycotoxin-concentration samples (1.6 μg/mL) was taken and placed in a 10 mL screw tube, where 4.5 mL of MRS culture medium was added. Tubes were incubated at 40 ± 1 °C for 24 h, with shaking. Subsequently, a subculture was grown in Petri dishes (9 cm diameter) with Lactobacilli MRS Agar (Difco BD, Sparks, MD, USA) using the streak plate technique and incubated at 40 ± 1 °C for 24 h in an aerobic atmosphere. Strains that presented a different morphological growth were grown in Petri dishes of 5.6 cm in diameter, and successive cultures were performed until pure cultures were obtained [21,32].

#### 2.3.2. Morphological Structure of Bacterial Isolates

To determine the morphological structure of the five isolates, stereoscopic, optical, and scanning electron microscopy (SEM) techniques were used. To determine bacterial colonial morphology, a sample was taken from a 12 h culture in Lactobacilli MRS Broth (Difco BD, Sparks, MD, USA), cultured in a 9 cm Petri dish with Lactobacilli MRS Agar and incubated at 40 ± 1 °C for 18 h. Observations were performed using a Velab VE-S7 stereoscopic microscope. The bacterial morphology was observed with a Nikon LABOPHOT-2 phase contrast microscope using phase 3, in order to clearly identify the bacterial cell wall structure. To determine the peptidoglycan layer traits, a Gram staining test was performed, following [39]. To assess the bacterial structures, samples were analyzed with SEM. Samples were prepared as follows: Bacterial strains were incubated at 40 ± 1 °C for 12 h, in Lactobacilli MRS broth with shaking. An aliquot of 1.5 mL was taken and placed in a 2 mL Eppendorf tube and centrifuged at 4 °C for 5 min at 4000 rpm in a Microfuge 22 R centrifuge (Beckman Coulter, Krefeld, Germany). The supernatant was decanted, the pellet was reconstituted in a 1.5 mL PBS (phosphate-buffered saline) solution and homogenized using a vortex, and 3 more washes with PBS were performed. The pellet was recovered, and 1.5 mL of glutaraldehyde solution (2.5% (*w*/*v*)) was added and left to stir at 120 rpm in an orbital shaker for 120 min. Three washes were then performed with PBS. The pellet was reconstituted in PBS (1:10 *w*/*v*) and left to stir at 40 ± 1 °C for 12 h, and then centrifuged at 4000 rpm for 5 min at 4 °C. The pellet was recovered, and successive washes were performed with 20, 50, 70, and 100% ethanol solutions [38]. Finally, the alcohol was evaporated at room temperature and placed in a sample holder. SEM observations were performed using a JEOL JSM-6380LV instrument, model 7582.

#### 2.3.3. Effect of the Bacterial Isolates on Mycotoxin Removal

To assess the effect of the isolates on mycotoxin removal, each isolate was placed in a 10 mL screw cap tube with 4.5 mL of MRS liquid medium and incubated at 40 ± 1 °C for 24 h under aerobic conditions with shaking. Then, 100 μL was taken, to which 1 μg mL^−1^ of each toxin was added, and the pH was adjusted to 7. The controls were the same as in Section 2.2.2. All tubes were incubated at 40 ± 1 °C for 24 h in a water bath on a shaking tray. Finally, extraction with ethyl acetate (1:3 *v*/*v*) was carried out for each mycotoxin sample, where the organic phase was recovered and evaporated by reduced dryness and stored until subsequent analyses. To quantify the residual mycotoxin in samples, HPLC-ESI-TOF-MS was used, following García-García et al. [36].

#### 2.3.4. Molecular Characterization of Isolates

To carry out DNA extraction, a DNeasy Plant Mini Kit (QIAGEN, Redwood City, CA, USA) was used. To amplify the 16S rDNA, a polymerase chain reaction (PCR) was carried out using the reverse and forward primers: 518f (5′ CCAGCAGCCGCGGTAATACG 3′), E334-16S-F (5′ CCRRACTCCTACGGGAGGCAGC 3′) and 928r (5′ CCGTCAATTCCTTTGAGTTT 3′), and U1115R (5′ AGGGTTGCGCTCGTTGCG 3′) [39]. The PCR was performed in a Thermal Cycler T100 (Bio-rad, Singapore), and the conditions were as follows: initial denaturation for 5 min at 95 °C, followed by 32 cycles of three steps: denaturation at 94 °C for 45 s, alignment at 56 °C for 45 s, and polimerization at 72 °C for 2 min. Reactions were finished with a 72 °C for 5 min cycle. For the reactions, the following conditions were used: sterile double-distilled water 20 µL; 2.5 µL of 10× buffer with Mg^2+^ in 15 mM dilution of MgCl_2_; 2 µL dNTP (diluted to 10 mM); 1 µL of primer f (in dilution to 10 mM); 1 µL of r primer (in dilution to 10 mM); 0.5 µL of Taq DNA polymerase (Invitrogen, Carlsbad, CA, USA) 5 units/µL; and 5 µL of DNA of the analyzed bacteria [40,41]. The purified PCR products were stored at −20 °C and sent for sequencing to the DNA Sequencing laboratory at UBIPRO FES-I, UNAM, Mexico. The obtained electropherograms were visualized and edited in Geneoius Prime version 2024.0.4. The sequences were compared with those in the National Center for Biotechnology Information (NCBI) database using the “BLASTN” tool (Basic Local Alignment Search Tool, Maryland, USA) [41,42,43]. Sequence similarities greater than 96%, when compared to sequences obtained from GenBank, were used as a criterion to indicate species identity. All accession numbers were obtained from GenBank. To construct the phylogenetic tree, a multiple sequence alignment was performed using Muscle on the data obtained from the 16S rDNA PCR sequencing, and the aligned sequences were used to construct a phylogenetic tree using Molecular Evolutionary Genetics Analysis (MEGA) version 11. To infer the evolutionary history of sequences, the neighbor–joining method was used, bootstrapping was performed for 1000 replicates, and the Tamura 3-parameter method was used to compute evolutionary distances [44,45]. The obtained tree was condensed with a cutoff value of 50.

### 2.4. Single Bacteria Characterization

#### 2.4.1. Biochemical Tests

To identify whether a strain was capable of metabolizing different carbohydrates, as well as the expression of some enzymes, a BBL Crystal GP test was used (Becton Dickinson and Company, Franklin Lakes, NJ, USA), and the strain was classified within groups following Bergey et al. (1994) and Barrow et al. (2003) [46,47].

#### 2.4.2. Biosorption Assay

To directly evaluate the role of the mycotoxin biosorption, individually and altogether, before the assays, the bacterium was cultured in MRS medium and incubated at 40 ± 1 °C for 24 h on a shaking water bath, sterilized for 20 min at 121 °C (1.02 atm), and centrifuged at 4000 rpm for 5 min, and then the pellet was recovered, washed three times, and resuspended in PBS buffer. Assays tubes were prepared with 1 µg mL^−1^ of each mycotoxin and the pH was adjusted to 7. Control 1 contained PBS buffer + mycotoxin, control 2 PBS buffer + inactive cells, and control 3 PBS buffer. All tubes were incubated at 40 ± 1 °C for 24 h with shaking in a water bath for 24 h. Finally, sample extraction was performed with ethyl acetate (1:3 *v/v*), the organic phase was recovered, and the latter was evaporated by the reduced dryness method. Samples were stored until analysis in HPLC-ESI-TOF-MS following García-García et al. [36].

### 2.5. Statistical Analysis

To assess differences in T-2 toxin, HT-2 toxin, and NEO concentration for the proventriculus bacterial consortium, we performed a one-way multiple analysis of variance (ANOVA), followed by Tuckey’s post hoc tests. Ordinary one-way ANOVA, followed by Sidak’s multiple comparisons test, were performed to compare the biosorption of the control and T-2, HT-2, and NEO, individually and together, in biosorption assays [48].

## 3. Results

### 3.1. Mycotoxin Production, Isolation, and Quantification

The mycelium of *F. sporotrichioides* resembled a whitish cotton and produced a reddish pigment. The presence of reproductive structures such as monophyalidic conidia (Figure 1A), isolated or paired chlamydospores (Figure 1B), and micro and macroconidia (Figure 1C) was observed. HPLC-ESI-TOF-MS analyses showed the presence of four type A trichothecenes (Figure 1D) that corresponded to the characteristic ions of DAS, T-2 toxin, HT-2 toxin, and NEO (Figure 1E). Type A trichothecenes were obtained in vitro, but the DAS concentration was extremely low (Table 1), so they were not included in further analyses in this study.

### 3.2. Bacterial Consortium

#### 3.2.1. Effect of Facultative Anaerobe Bacteria Consortium on T-2 Mycotoxin Removal

The bacterial consortium removed T-2 toxin by 16.27% and 25.41%, P1 and P2, respectively. Differences between groups were found (ANOVA *p* = 0.014; F = 9.455; Figure 2A). Tukey’s post hoc tests showed significant differences when comparing P2 to C (*p* < 0.012; q = 6.07; DF = 6); and no differences were found when comparing P1 and C (*p* < 0.074; q = 3.88; DF = 6), and P1 to P2 (*p* = 0.34; q =2.18; DF = 6).

#### 3.2.2. Effect of Facultative Anaerobe Bacteria Consortium on HT-2 Mycotoxin Removal

The bacterial consortium removed HT-2 toxin by 20.01% and 25.91%, P1 and P2, respectively. Differences between groups were found (ANOVA *p* = 0.008; F = 11.74; Figure 2B). Tukey post hoc tests showed significant differences when comparing both P1 (*p* = 0.03; q = 5.05; DF = 6) and P2 (*p* = 0.009; q = 6.54; DF = 6) to C, and no differences were found when comparing P1 and P2 (*p* = 0.57; q =1.49; DF = 6).

#### 3.2.3. Effect of Facultative Anaerobe Bacteria Consortium on NEO Mycotoxin Removal

The bacterial consortium removed NEO by 63.53% and 65.57%, P1 and P2, respectively. Differences between groups were found (ANOVA *p* = 0.008; F = 11.74; Figure 2C). Tukey post hoc tests showed significant differences when comparing both P1 (*p* = 0.03; q = 5.05; DF = 6) and P2 (*p* = 0.009; q = 6.54; DF = 6) to C; and no differences were found when comparing P1 and P2 (*p* = 0.57; q = 1.49; DF = 6).

### 3.3. Bacterial Isolates

#### 3.3.1. Morphological Structure of Bacterial Isolates

The bacterial colonies’ size ranged from small to large, and three pigmentation types, transparent, white, and cream-colored, were present (Figure 3). Colonies presented either regular or irregular shapes and entire edges with convex and irregular elevation. All isolates were bacillus-shaped (Figure 4) and Gram-positive. SEM analyses allowed accurately determining the bacteria shape, where bacilli of different sizes were identified, I5 (2.81 nm) and I2 (2.08 nm) had a greater length, while bacteria I3 (1.81 nm), I4 (1.87 nm) and I1 (1.73 nm) were smaller. The presence of endospores was only found in I5 (Figure 5).

#### 3.3.2. Molecular Characterization of Isolates

According to the 16S rDNA PCR results, I1 presented 1069 bp (base pairs); I2, 1501 bp; I3, 1459 bp; I4, 1096 bp; and I5, 1544 bp. When the sequences were compared to the NIST database, the nucleotide identity was greater than 95% in all isolates. The identity of the isolates corresponded to *Lactiplantibacillus plantarum* (I1 and I3), *Bacillus* sp. (I2), *Limosilactobacillus reuteri* (I4), and *Bacillus cereus* (I5; Table 2). The 16S rDNA gene sequences of the five isolates were deposited in the GenBank database under the accession numbers: I1 (PQ301253), I2 (PQ301254), I3 (PQ301255), I4 (PQ301256), and I5 (PQ301257).

The 16S rDNA gene sequence was successfully aligned and compared with 21 known sequences obtained from GenBank. *Clostridium perfringens* (FN356962.1) was used as the outgroup. Two main groups were distinguished: the *Bacillus* group in the upper part, and the *Lactobacillus* group at the bottom of the tree (Figure 6). Strains I1 and I3 were clustered together with a bootstrap value of 93% close to the cluster containing *Lactiplantibacillus plantarum* (MF541084 and MT473659) and with a bootstrap value of 47%. Strain I2 was clustered with *Bacillus thuringiensis* (LC799634.1), with a bootstrap value of 49%. Strain I4 was clustered with *Limosilactobacillus reuteri* (PQ013831), with a bootstrap value of 100%. Strain I5 was clustered with *Bacillus cereus* (NR074540.1), with a bootstrap value of 80%.

#### 3.3.3. Effect of Bacterial Isolates on Mycotoxin Concentration

T-2. While I3, I4, and I5 reduced T-2 toxin concentration (26.52%, 18.70%, and 28.35%; Figure 7) when compared to C; I1 and I2 incremented the T-2 toxin concentration compared to C.HT-2. I1, I3, I4, and I5 reduced the HT-2 toxin concentration (28.27%, 1.99%, 1.38%, and 32.84%) when compared to C. I2 slightly incremented the HT-2 toxin compared to C (Figure 7).NEO. I5 was the only strain that reduced the NEO concentration by 27.14% compared to C. In contrast, the rest of the isolates increased the NEO concentration when compared to C (Figure 7).

Based on the results obtained in this section, given that *B. cereus* (I5) was the only bacteria capable of reducing the concentration of the three studied mycotoxins in greater proportions, it was selected to carry out the subsequent experiments.

### 3.4. Single Bacteria

#### 3.4.1. Biochemical Tests

The BBL crystal test results showed positive for arabinose, fructose, *p*-nitrophenyl-β-d-glucoside, and escualine (Table 3), according to Bergey manual [46] and Cowan and Steel (2003) [47]. These results match the *Bacillus* Senso lacto group, which includes *B. cereus*, *B. anthracis,* and *B. thuringiensis*.

#### 3.4.2. Biosorption Assay

*B. cereus* individually biosorbed T-2 by 24.41 ± 2.18% compared to C (*p* ≤ 0.0001; t = 14.43; df =12; Figure 8A); HT-2 by 11.69 ± 1.85% compared to C (*p* ≤ 0.0001; t = 7.60; df =12); and NEO by 18.63 ± 3.36% compared to C (*p* ≤ 0.0001; t = 13.43; df =12).

When the biosorption was tested with the three mycotoxins simultaneously, *B. cereus* significantly biosorbed all types, with T-2 by 7.39 ± 2.01%, HT-2 by 10.33 ± 0.98%, and NEO by 14.96 ± 2.14%, and differences between groups were found (ANOVA *p* ≤ 0.0001; F = 159.5; Figure 8B). Sidak’s multiple comparisons test showed significant differences between C and T-2 (0.0001; t = 6.398; df = 12), C and HT-2 (<0.0001; t = 9.825; df = 12), and C and NEO (<0.0001; t = 15.79; df = 12).

## 4. Discussion

In recent years, the incidence of type A trichothecenes has increased in grains for food production, such as corn, rice, wheat, sorghum, and oats, among others [14]. Although T-2 toxin has been detected to the greatest extent, as type A trichothecenes are part of the same biosynthetic pathway, they are often found to cause multi-mycotoxicosis [49], and a synergistic effect between T-2 toxin and its metabolites has been further recognized [50,51]. Therefore, proposing biological methods that remove more than one type of A trichothecene mycotoxin is still of great scientific importance. Here, we found that (i) all type A trichothecenes were produced by *F. sporotrichioides* at different concentrations; (ii) the proventriculus bacterial consortium degraded all tested trichothecenes, T-2 (>16%), HT-2 (>20%), and NEO (>60%); (iii) *B. cereus* was the only isolate found to reduce the three type A trichothecenes here studied; (iv) *B. cereus* biosorption accounted for the 86.10% of the total removal in T-2, 35.59% in HT-2, and 68.64% in NEO.

Our results showed the biosynthesis of the four type A trichothecenes from *F. sporotrichioides*, differing in their concentration. *F. sporotrichioides* has been reported to consistently produce type A trichothecenes and has been recognized as the main source for T-2 and HT-2 [9,52]. In the type A trichothecene biosynthesis pathway, DAS is initially synthesized as the *F. sporotrichioides* mycelium grows and develops, depending on the abiotic conditions. Then, T-2 and NEO are the main secondary metabolites [53,54]. In addition, the presence of T-2 in media causes the fungus to adsorb and hydrolyze the acetyl C-4, therefore increasing HT-2 [55,56]. Accordingly, in our results, quantitative differences in the production of type A trichothecenes were documented between *F. sporotrichioides* and *F. langsethiae*, where both produced higher amounts of T-2, but *F. sporotrichioides* was more prolific in producing HT-2, and *F. langsethiae* produced higher amounts of DAS and NEO [57]. Moreover, the production of trichothecenes is influenced by other factors such as temperature, light exposure, and stress [58].

Here, we found that the proventriculus bacterial consortium reduced the concentration of T-2, HT-2, and NEO. The capacity of the facultative anaerobe consortia to act as mycotoxin inactivators has been widely recognized and reviewed in recent studies [18,59,60,61]. The alkaline pH in a broiler’s intestine has been found to enhance the enzymatic activity of the microorganisms that catalyze mycotoxins [62]. Although the main function of the proventriculus consists in gastric juice secretion to act in the gizzard and the first duodenum portion, the presence of a diverse group of bacteria [63] capable of producing enzymes to remove the acetyl group in T-2 has been described [64]. According to our findings, a microbial consortium from a broiler’s large intestine was found to remove 12 type A and B trichothecenes, including DAS, T-2, HT-2, and NEO [21].

In this study, based on the results obtained in the phylogenetic tree, *Bacillus* (*B. thuringiensis* and *B. cereus*) and *Lactobacillus* (*L. plantarum* and *L. reuteri*) species were found to differently remove type A trichothecenes. Accordingly, several species of bacteria have been recognized to have the ability to remove mycotoxins, including the *Bacillus* and *Lactobacillus* genera [65,66,67,68]. Moreover, the intestinal microbiota composition in birds fed with an immobilized diet has recently been linked to an insufficient stress response to external stimuli, resulting in a significant increase in the abundance of pathogenic groups in the intestinal bacterial community [69]. Accordingly, both *Bacillus* species found in this study, *B. thuringiensis* and *B. cereus*, are considered pathogens [70,71]. In this study, *B. cereus* was found, for the first time, to reduce the concentration of the three studied trichothecenes, T-2 toxin, HT-2 toxin, and NEO, and has previously been recognized as causing foodborne diseases and local and systemic infections in humans and animals [72,73]. The presence of *B. cereus* in broiler proventriculus could be due to its ability to produce endospores, since this structure is heat- and dehydration-resistant, allowing it to temporally associate with the gastrointestinal microbiome [74], and its presence has also been reported in poultry feed and broiler production sites [75], and as a form of contamination, due to broiler waste ingestion in their reproductive system [76]. In contrast, the use of *Bacillus* endospores as probiotics for animals and humans has been widely documented [77], and in bacteria within the *B. cereus* group (*Bacillus* Senso lacto group, including *B. cereus*, *B. anthracis*, and *B. thuringiensis*), the absence of enterotoxins synthesis genes has been reported [78], thus promoting competitive exclusion within the microbiota, immune exclusion and adhesion site competition, and the production of antimicrobial agents [79]. Therefore, given that the gene expression responsible for enterotoxin synthesis depends on factors such as pH and culture media [80], in future studies, it is suggested to carry out experiments that identify the presence of enterotoxins in culture media, and to consider the study of *B. cereus* enzymatic components instead of contemplating the use of the entire microbe as a biological control for mycotoxins in living organisms, food, and feed. Moreover, the capacity of the species found here to remove other mycotoxins has been previously documented: Fumonisins by *B. thuringiensis* [81], aflatoxin B1 and T-2 by *B. cereus* [82,83], aflatoxins and zearalenone (ZEN) by *L. plantarum* [18,84], and ZEN by *L. reuteri* [84,85]. Differences in the trichothecene removal assays as obtained in this study, between *L. plantarum* in I1 and I3, further suggest the presence of different strains with different removal capacities.

Here, we found that *B. cereus* biosorption accounted for 86.10% of the total removal of T-2, 35.59% of HT-2, and 68.64% of NEO. Since whole cells were used in this experiment, biosorption was attributed to the cell wall. Biosorption involves a rapid and direct binding of the toxin capable of being released, depending on the bacterial affinity towards the toxin [86]. Microbial cells act as organic adsorbents by binding mycotoxins to their surface [17]. Cell wall composition is highly diverse in physical structure, chemical components, and interactions among microbe species, and therefore in adsorption capacity [87,88]; however, the peptidoglycans and polysaccharides in the cell wall are responsible for mycotoxin adsorption [41,89,90]. Gram-positive bacteria form a thick peptidoglycan multi-layer wall, with polysaccharides, proteins, lipoteichoic, and teichoic acids on their surface [91], causing their characteristic hydrophobicity and allowing them to act as electron donors and weak electron acceptors by generating Lewis acid–base interactions and Van der Walls forces between the proteins in their S layer (outer cell wall layer) and the functional groups present in trichothecenes structures, therefore leading to a decrease in their concentration [29]. The S layer of *B. cereus* is mainly composed of glucose, N-acetylgalactosamine, N-acetyl-mannosamine, and N-acetyl-glucosamine glycoproteins with either oblique, squared, or hexagonal reticular symmetries, exposing positively charged residues on the surface [92,93]. In this study, although the cell wall of *B. cereus* biosorbed the three tested mycotoxins, it mostly contributed to NEO biosorption. We suggest that since NEO’s two functional groups are acetyls in C-4 and C-6, and considering its structure and smaller size, it allows a greater interaction with the *B. cereus* cell wall compared to T-2 and HT-2. In the case of HT-2, its isovaleric acid at C-8 could reduce the interaction with the *B. cereus* cell wall, since its O in C-17 is less exposed. Similarly, in T-2, the presence of isovaleric acid at C-8, due to its conformational arrangement, may negatively influence biosorption [9,30,68]. Considering the results obtained in the assay containing the three trichothecenes, competitive adsorption could be occurring among mycotoxins, since their structure only differs in their C-4, C-6, and C-8 functional groups, NEO being the most competitive due to its acetyl groups at C-4 and C-6; however, more in-depth studies are required to verify this hypothesis. Similar biosorption percentages as those found in this study have previously been found using lactic acid bacterial species, such as *Lactococcus lactis,* which was able to biosorb T-2 by 31% [94], and another *Lactobacillus* species was found to biosorb T-2 by 52% [95].

Moreover, the results showed that when the bacterial isolates were tested for removal of T-2, HT-2, and NEO, in some cases, the trichothecene concentration instead increased. This increase in mycotoxins concentration could be due to *F. sporotrichiodes* assimilating and metabolizing the mycotoxins to a glycosidic trichothecene, when coming into contact with toxins within the media. Glycosidic conjugation reactions are recognized as a metabolic pathway used by filamentous fungi, playing an important role in self-defense against other mycotoxin-producing fungi [96]. This kind of biotransformation consists in adding a carbohydrate, generally glucose, an amino acid, or a short lipid chain [97]. Particularly for trichothecenes, the glycosylation of T-2 and HT-2 has been recognized in phase two of the xenobiotic metabolism in the genus *Fusarium,* to avoid damage due to trichothecenes [98,99]. Furthermore, in the case of NEO, since its structure is similar to deoxynivalenol, it can be biotransformed by binding to a glycoside [100]. Accordingly, bacteria have been documented to express glycosyltransferases that may be involved in the glycosylation of several compounds; thus, the fraction of T-2, HT-2, and NEO could be masked by conjugation [101], causing an increase in mycotoxin concentration, as found here.

## 5. Conclusions

Overall, this study highlights the relevance of the proventriculus anaerobe bacteria as a control alternative to type A trichothecene contamination in poultry health and food safety. This study is the first to document *B. cereus* potential as an effective multi-type A trichothecene detoxifier and highlights the importance of developing enzymatic studies.

## Figures and Tables

**Figure 1 microorganisms-12-02236-f001:**
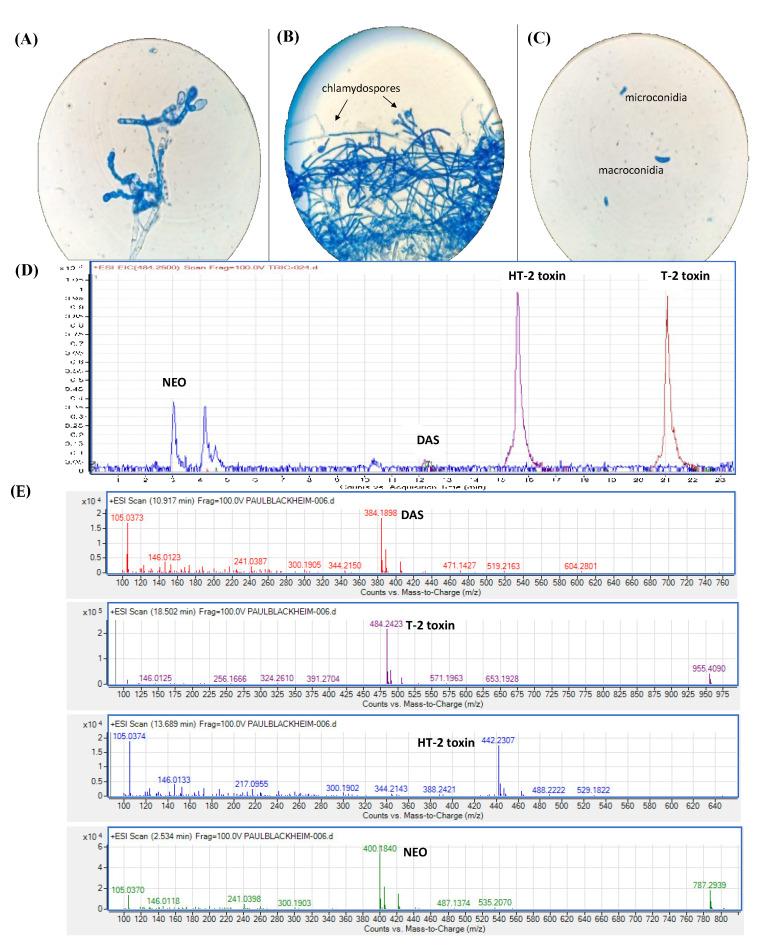
Reproductive structures of *Fusarium sporotrichioides* and in vitro production of type A trichothecenes. (**A**) phyalidic conidia (1000×); (**B**) chlamydospores (400×); (**C**) micro and macroconidia (1000×); (**D**) type A trichothecenes shown by HPLC-ESI-TOF-MS; (**E**) characteristic ions of (DAS), T-2 toxin, HT-2 toxin, and (NEO). DAS = diacetoxyscirpenol; T-2 = T-2 toxin; HT-2 = HT-2 toxin; NEO = neosolaniol.

**Figure 2 microorganisms-12-02236-f002:**
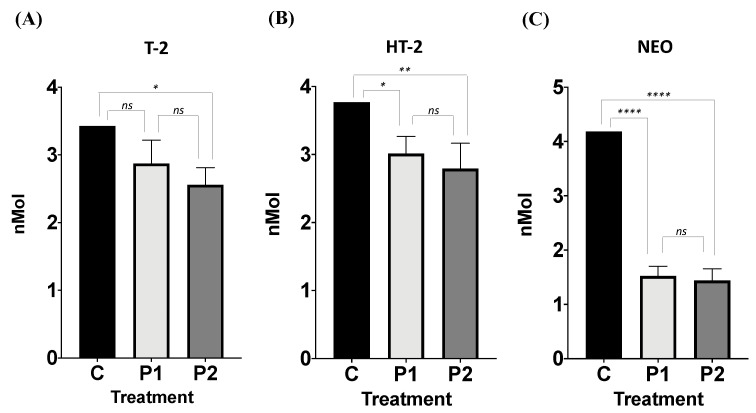
Effect of facultative anaerobe bacterial consortium from broiler proventriculus on trichothecene mycotoxin removal (nMol). C = control tube, P1, P2 = experimental groups; (**A**) T-2 = T-2 toxin; (**B**) HT-2 = HT-2 toxin; (**C**) NEO = neosolaniol. (ns, non-significant; * *p* < 0.05; ** *p* < 0.01; **** *p* < 0.0001).

**Figure 3 microorganisms-12-02236-f003:**
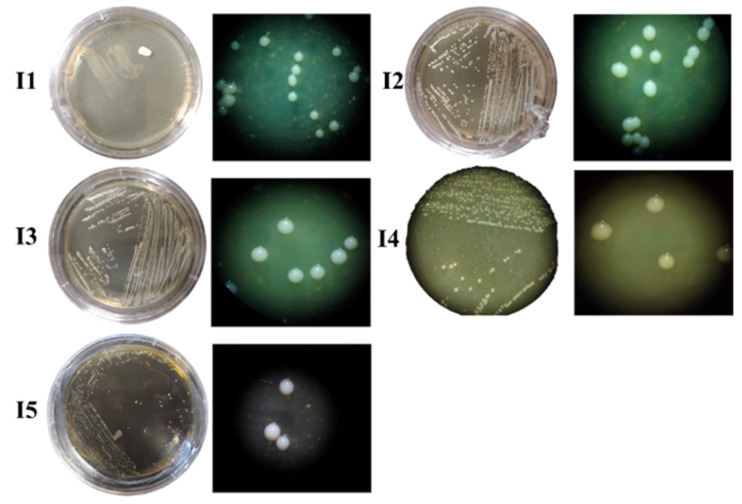
Bacterial isolate (I1, I2, I3, I4, and I5) micrographs from the facultative anaerobic consortium coming from a broiler’s proventriculus (20×; stereoscopic microscope).

**Figure 4 microorganisms-12-02236-f004:**
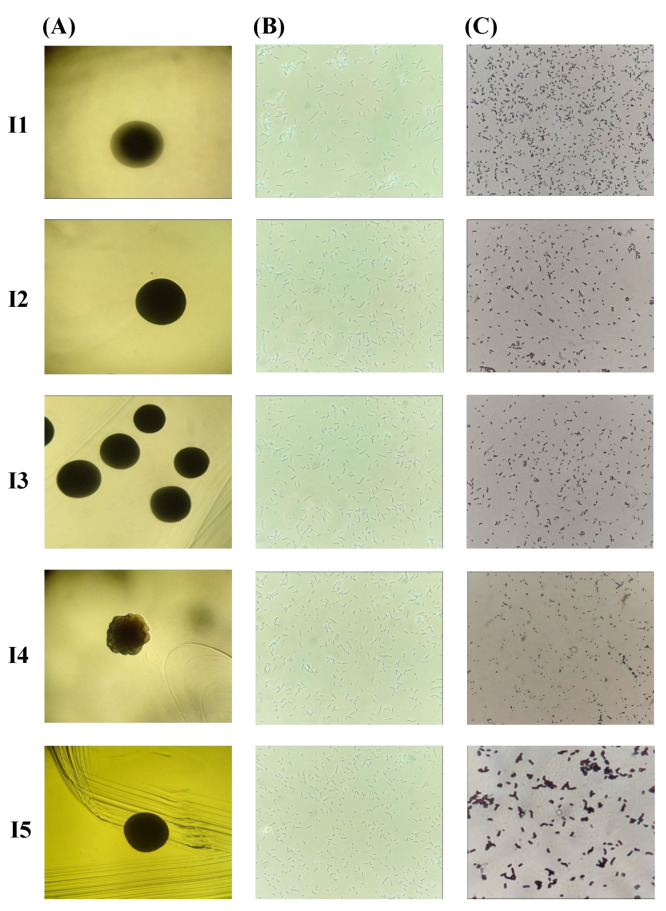
Micrographs showing the morphology of the colonial and bacterial isolates (I1, I2, I3, I4, and I5). (**A**) colony morphology; (**B**) shape; (**C**) Gram staining test. edge (40×; phase-contrast microscope; phase 0); shape (400×; phase-contrast microscope; phase 3); Gram staining test (400×; phase-contrast microscope; phase 0).

**Figure 5 microorganisms-12-02236-f005:**
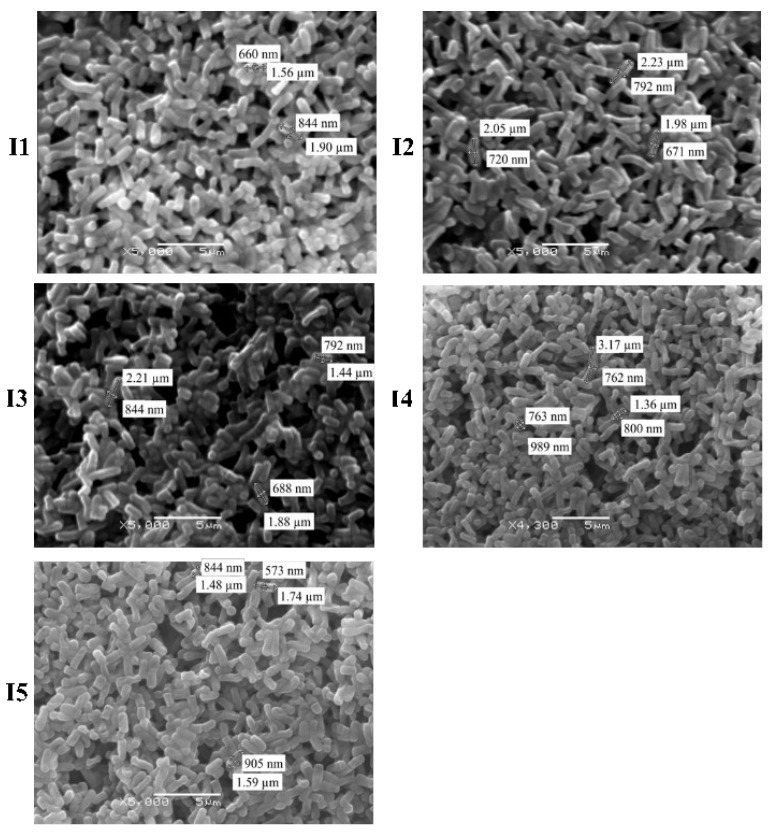
Micrographs showing size and length of bacterial isolates (I1, I2, I3, I4 and I5; 5000×; scanning electron microscopy SEM).

**Figure 6 microorganisms-12-02236-f006:**
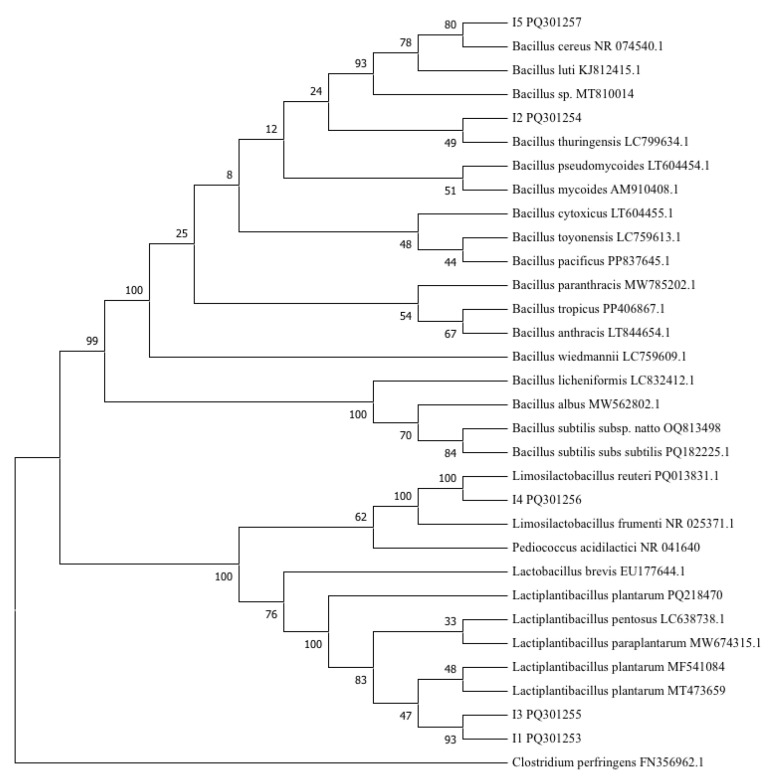
Phylogenetic tree based on 16S rDNA gene sequence analysis of facultative anaerobe bacterial isolates. Numbers next to nodes indicate posterior probability.

**Figure 7 microorganisms-12-02236-f007:**
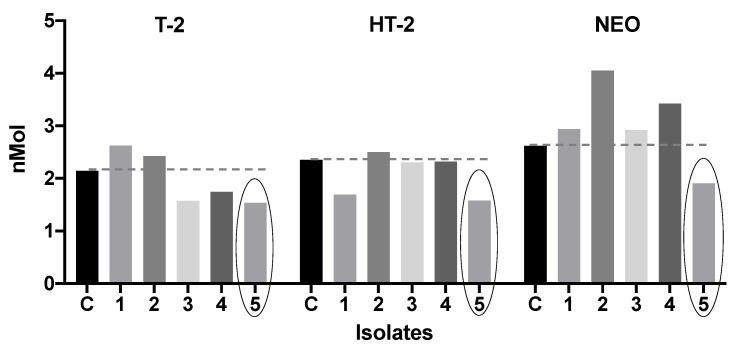
Effect of bacteria isolated from broiler’s proventriculus on T-2, HT-2, and NEO trichothecene concentration (nMol). The dotted line represents the level of the control tube. The circle indicates the bacteria that most reduced all trichothecenes. C = control; 1 = *Lactiplantibacillus plantarum*; 2 = *Bacillus* sp.; 3 = *Lactiplantibacillus plantarum*; 4 = *Limosilactobacillus reuteri*; 5 = *Bacillus cereus*. T-2 = T-2 toxin; HT-2 = HT-2 toxin; NEO = neosolaniol.

**Figure 8 microorganisms-12-02236-f008:**
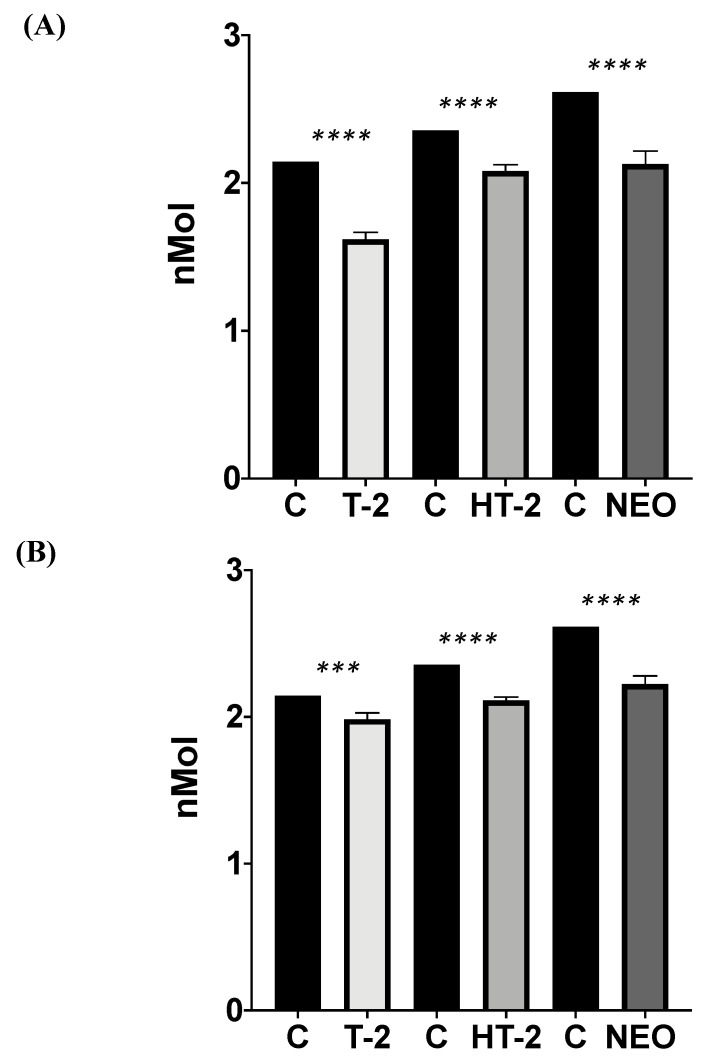
*Bacillus cereus* biosorption assays of T-2, HT-2, and NEO type A trichothecenes mycotoxins when tested individually (**A**) and altogether (**B**). C = control; T-2 = T-2 toxin; HT-2 = HT-2 toxin; NEO = neosolaniol. (*** *p* = 0.0001; **** *p* ≤ 0.0001).

**Table 1 microorganisms-12-02236-t001:** Type A trichothecene content (µg/mL) as obtained in *Fusarium sporotrichioides*.

Type A Trichothecenes Concentration				
Extract	DAS (µg/mL)	T-2 (µg/mL)	HT-2 (µg/mL)	NEO (µg/mL)
1	1.35	2.39	1.87	67.62
2	0	257.59	1134.63	79.17
3	36.84	1351.49	16.35	65.32

Culture age: extract 1 = 77 days, extract 2 = 21 days, and extract 3 = 6 days. DAS = diacetoxyscirpenol; T-2 = T-2 toxin; HT-2 = HT-2 toxin; NEO = neosolaniol.

**Table 2 microorganisms-12-02236-t002:** Taxonomic identification of broiler proventriculus bacterial isolates by PCR analysis.

I	Nearest Matched Species from GenBank	E-Value	Query-Cover (%)	Identity (%)	GenBank Accession Number
1	*Lactiplantibacillus plantarum* PP16	0	99	99.61	MF541084.1
2	*Bacillus* sp.	0	100	99.58	MT810014.1
3	*Lactiplantibacillus plantarum* IMAU98304	0	95	95.29	MT473659.1
4	*Limosilactobacillus reuteri* V3266	0	99	99.52	PQ013831
5	*Bacillus cereus* DJ29	0	100	100	CP162505

(I) isolates.

**Table 3 microorganisms-12-02236-t003:** *Bacillus cereus* capacity to metabolize carbohydrate types and enzyme expression. Positive (+); negative(−).

Test	Result
Trehalose	−
Sucrose	−
Arabinose	+
*p*-nitrophenyl-β-d-glucoside	+
*p*-nitrophenyl-phosphate	−
Urea	−
Lactose	−
Mannitol	−
Glycerol	−
*p*-nitrophenyl-β-d-cellobioside	−
*p*-nitrophenyl-α-d-maltoside	−
Escualine	+
Methyl-α and -β glucoside	−
Maltotriose	−
Fructose	+
Proline and Leucine-*p*-nitroanilide	−
*o*-nitrophenyl-β-d-galactoside and *p*-nitrophenyl-α-d-galactoside	−
Arginine	−

## Data Availability

All data generated or analyzed during this study are included in the published article, and the 16S rDNA gene sequences of isolates have been deposited in the GenBank database under their accession numbers (PQ301253 to PQ301257).

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
