# Peer review of "Study of Bacillus cereus as an Effective Multi-Type A Trichothecene Inactivator"

_microorganisms, 2024, doi:10.3390/microorganisms12112236_

Round 1
Reviewer 1 Report
Comments and Suggestions for Authors
The authors have submitted a very detailed article, which mainly reports B. cereus potentially as an effective multi-type A 501 trichothecene inactivator. Type A trichothecenes are common contaminants in cereal grain and due to their high toxicity, have been considered of greatest concern to food and feed safety. To mitigate the toxicity, this study developed a biological strategy, using some facultative anaerobe bacteria isolated from the broiler’s proventriculus. The 5 most representative bacterial strains were isolated and purified, their morphological structure characterized, their 16S rDNA sequence analyzed to assess their taxonomic identification, and isolates removal assays were performed. Considering the toxicity of multi-type A 501 trichothecene, the study is very meaningful, and the research is very rich in data.
I think the manuscript can be accepted.
Besides, some keywords need to be deleted, for example, Biosorption and 16S rDNA sequencing.
Author Response
RESPONSE TO REVIEWER 1
Manuscript 3279645 entitled "Study of Bacillus cereus as an effective multi-type A trichothecene inactivator" submitted to Microorganisms.
We really appreciate the reviewers for their advisable and objective comments about our manuscript. All of them will be included in the latest version and improves the readability and interests of the readers of the journal. In essence, we tried to deeply meet all your requests. However, we are fully available to attend any additional points that does not meet with your expectations in depth. In the following section, each comment is listed and followed by our response. The remarks made by the reviewers are written in bold face, and our respective answers are included in italics.
Comments to authors
Reviewer 1
The authors have submitted a very detailed article, which mainly reports B. cereus potentially as an effective multi-type A trichothecene inactivator. Type A trichothecenes are common contaminants in cereal grain and due to their high toxicity, have been considered of greatest concern to food and feed safety. To mitigate the toxicity, this study developed a biological strategy, using some facultative anaerobe bacteria isolated from the broiler’s proventriculus. The 5 most representative bacterial strains were isolated and purified, their morphological structure characterized, their 16S rDNA sequence analyzed to assess their taxonomic identification, and isolates removal assays were performed. Considering the toxicity of multi-type A trichothecene, the study is very meaningful, and the research is very rich in data. I think the manuscript can be accepted.
Comment 1. Besides, some keywords need to be deleted, for example, Biosorption and 16S rDNA sequencing.
Response
We thank for the recommendation. As suggested by the reviewer, biosorption and 16S rDNA sequencing were deleted in the keywords section. Now it reads: “Keywords: Biological mycotoxin decontamination; Diacetoxyscirpenol (DAS); HT-2 toxin; Multi-mycotoxicosis; Neosolaniol (NEO); Proventriculus bacterium; T-2 toxin; Type A trichothecenes.”
Reviewer 2 Report
Comments and Suggestions for Authors
The authors present a thorough study of the removal of trichothecene by Bacillus cereus. The data is sound and I recommend publication after a minor revision.
The material and methods offer a clear understanding of experiments.
The figures show the data in variety of formats.
The statistical analyses are sound.
While the discussion is thorough and presents results based on the data in a rigorous way, the conclusion was a bit underwhelming. Please expand upon your discussion in your conclusion in a meaningful way.
Author Response
RESPONSE TO REVIEWER 2
Manuscript 3279645 entitled "Study of Bacillus cereus as an effective multi-type A trichothecene inactivator" submitted to Microorganisms.
We really appreciate the reviewers for their advisable and objective comments about our manuscript. All of them will be included in the latest version and improves the readability and interests of the readers of the journal. In essence, we tried to deeply meet all your requests. However, we are fully available to attend any additional points that does not meet with your expectations in depth. In the following section, each comment is listed and followed by our response. The remarks made by the reviewers are written in bold face, and our respective answers are included in italics.
Comments to authors
Reviewer 2
The authors present a thorough study of the removal of trichothecene by Bacillus cereus. The data is sound and I recommend publication after a minor revision. The material and methods offer a clear understanding of experiments. The figures show the data in variety of formats. The statistical analyses are sound.
Comment 1. While the discussion is thorough and presents results based on the data in a rigorous way, the conclusion was a bit underwhelming. Please expand upon your discussion in your conclusion in a meaningful way.
Response
Thank you for pointing this out, as requested, in this new version of the manuscript, the conclusion is significantly expanded trying to highlight the most relevant aspects of the study. “Overall, this study highlights the relevance of the proventriculus anaerobe bacteria as a control alternative to type A trichothecene contamination in poultry health and food safety. This study firstly documented B. cereus potential as an effective multi-type A trichothecene detoxifier and highlights the importance of developing enzymatic studies.”
Reviewer 3 Report
Comments and Suggestions for Authors
Regarding Manuscript ID: microorganisms-3279645
Type of manuscript: Article
Title: Study of Bacillus cereus as an effective multi-type A trichothecene inactivator
Authors: Fernando Abiram Garcia-Garcia, Eliseo Cristiani-Urbina *, Liliana Morales-Barrera, Olga Nelly Rodríguez-Peña, Luis Barbo Hernández-Portilla, Jorge Eduardo Campos Contreras, Cesar Mateo Flores-Ortiz *
To Authors
The authors used different bacterial strains of Bacillus and Lactobacillus isolated from broiler’s proventriculus to simultaneous inactivate mycotoxins like type A trichothecenes that could affect cereal grains.
The quantitatively predominant mycotoxins secreted by Fusarium sporotrichioides were T-2 toxin (T-2), HT-2 toxin (HT-2) and neosolaniol (NEO), while the most efficient strain was Bacillus cereus, alone or in synergism with Lactobacillus species.
Overall, it is a complex well-structured work, suitable for the purpose, which takes into account the deepening of the studies at the molecular/enzymatic level.
I invite the authors to reconsider some aspects.
Q1: I am not sure that the phrase “Control 1 contained MRS medium + mycotoxin, control 2 MRS medium + live cells, and control 3 MRS medium. Since not changes in controls 2 and 3 were detected, only the result of control 1 was used in the analysis.” should be appear in the both sections of the manuscript (2.3.3 Effect of the bacterial isolates on mycotoxin removal and2.4.2 Biosorption assay).
Q2: Maybe the authors should revise the division of paragraph 3.2. Bacterial Consortium into subsections:
3.2.1 Effect of facultative anaerobe bacteria consortium on T-2 mycotoxin removal.
3.2.2 Effect of facultative anaerobe bacteria consortium on HT-2 mycotoxin removal.
3.2.3 Effect of facultative anaerobe bacteria consortium on NEO mycotoxin removal.
Q3: Please revise the followings:
3.3.3. Effect of bacterial isolates on mycotoxin concentration
● T-2. While I3, I4 and I5 reduced T-2 toxin concentration (26.52%, 18.70%, and 28.35%; Fig. 7) when compared to C; I1 and I3 incremented T-2 toxin concentration compared to C.
Author Response
RESPONSE TO REVIEWER 3
Manuscript 3279645 entitled "Study of Bacillus cereus as an effective multi-type A trichothecene inactivator" submitted to Microorganisms.
We really appreciate the reviewers for their advisable and objective comments about our manuscript. All of them will be included in the latest version and improves the readability and interests of the readers of the journal. In essence, we tried to deeply meet all your requests. However, we are fully available to attend any additional points that does not meet with your expectations in depth. In the following section, each comment is listed and followed by our response. The remarks made by the reviewers are written in bold face, and our respective answers are included in italics.
Comments to authors
Reviewer 3
The authors used different bacterial strains of Bacillus and Lactobacillus isolated from broiler’s proventriculus to simultaneous inactivate mycotoxins like type A trichothecenes that could affect cereal grains. The quantitatively predominant mycotoxins secreted by Fusarium sporotrichioides were T-2 toxin (T-2), HT-2 toxin (HT-2) and neosolaniol (NEO), while the most efficient strain was Bacillus cereus, alone or in synergism with Lactobacillus species. Overall, it is a complex well-structured work, suitable for the purpose, which takes into account the deepening of the studies at the molecular/enzymatic level. I invite the authors to reconsider some aspects.
Comment 1. I am not sure that the phrase “Control 1 contained MRS medium + mycotoxin, control 2 MRS medium + live cells, and control 3 MRS medium. Since not changes in controls 2 and 3 were detected, only the result of control 1 was used in the analysis.” should be appear in both sections of the manuscript (2.3.3 Effect of the bacterial isolates on mycotoxin removal and 2.4.2 Biosorption assay).
Response
In sections 2.2.2, 2.3.3 and 2.4.2, three controls were included, however, their identity was the same for 2.2.2 and 2.3.3, and different for 2.4.2. In all cases, controls 2 and 3 were not included in the results since no changes were detected. To avoid misunderstandings, in the new version of the manuscript we paraphrase what refers to controls in order to be clearer. Now it reads:
In 2.2.2 Enrichment of the bacterial consortium and removal of mycotoxins
“Control 1 contained MRS medium + mycotoxin, control 2 MRS medium + live cells, and control 3 MRS medium. Since no differences in controls 2 and 3 were detected, only the control 1 was included in the analyzes throughout the manuscript.”
In 2.3.3 Effect of the bacterial isolates on mycotoxin removal
“The controls were the same as in section 2.2.2.”
In 2.4.2 Biosorption assay
“Control 1 contained PBS buffer + mycotoxin, control 2 PBS buffer + inactive cells, and control 3 PBS buffer.”
Comment 2. Maybe the authors should revise the division of paragraph 3.2. Bacterial Consortium into subsections:
3.2.1 Effect of facultative anaerobe bacteria consortium on T-2 mycotoxin removal.
3.2.2 Effect of facultative anaerobe bacteria consortium on HT-2 mycotoxin removal.
3.2.3 Effect of facultative anaerobe bacteria consortium on NEO mycotoxin removal.
Response
We agree with this comment, therefore, subsections for each mycotoxin were added. Now it reads.
“3.2. Bacterial Consortium
3.2.1 Effect of facultative anaerobe bacteria consortium on T-2 mycotoxin removal.
The bacterial consortium removed T-2 toxin in 16.27% and 25.41%, P1 and P2, respectively. Differences between groups were found (ANOVA p = 0.014; F = 9.455; Fig. 2-A). Tukey’s post-hoc tests showed significant differences when compared P2 to C (p < 0.012; q = 6.07; DF = 6); and no differences were found when compared P1 and C (p < 0.074; q = 3.88; DF = 6) and P1 to P2 (p = 0.34; q =2.18; DF = 6).
3.2.2 Effect of facultative anaerobe bacteria consortium on HT-2 mycotoxin removal.
The bacterial consortium removed HT-2 toxin in 20.01% and 25.91%, P1 and P2, respectively. Differences between groups were found (ANOVA p = 0.008; F = 11.74; Fig. 2-B). Tukey’s post-hoc tests showed significant differences comparing both P1 (p = 0.03; q = 5.05; DF = 6) and P2 (p = 0.009; q = 6.54; DF = 6) to C, and no differences were found when compared P1 and P2 (p = 0.57; q =1.49; DF = 6).
3.2.3 Effect of facultative anaerobe bacteria consortium on NEO mycotoxin removal.
The bacterial consortium removed NEO in 63.53% and 65.57%, P1 and P2, respectively. Differences between groups were found (ANOVA p = 0.008; F = 11.74; Fig. 2-C). Tukey post-hoc tests showed significant differences when compared both P1 (p = 0.03; q = 5.05; DF = 6) and P2 (p = 0.009; q = 6.54; DF = 6) to C; and no differences were found when compared P1 and P2 (p = 0.57; q = 1.49; DF = 6).”
Comment 3. Please revise the followings: 3.3.3. Effect of bacterial isolates on mycotoxin concentration T-2. While I3, I4 and I5 reduced T-2 toxin concentration (26.52%, 18.70%, and 28.35%; Fig. 7) when compared to C; I1 and I3 incremented T-2 toxin concentration compared to C.
Response
We appreciate this comment, since when we reviewed it in detail, we realized that we had repeated I3, and in the latest version of the manuscript it was corrected. Now it reads: “T-2. While I3, I4 and I5 reduced T-2 toxin concentration (26.52%, 18.70%, and 28.35%; Fig. 7) when compared to C; I1 and I2 incremented T-2 toxin concentration compared to C.”